# Exact Subgraph Isomorphism Network for Predictive Graph Mining

## Abstract

In the graph-level prediction task (predict a label for a given graph), the information contained in subgraphs of the input graph plays a key role. In this paper, we propose Exact subgraph Isomorphism Network (EIN), which combines the exact subgraph enumeration, a neural network, and a sparse regularization. In general, building a graph-level prediction model achieving high discriminative ability along with interpretability is still a challenging problem. Our combination of the subgraph enumeration and neural network contributes to high discriminative ability about the subgraph structure of the input graph. Further, the sparse regularization in EIN enables us 1) to derive an effective pruning strategy that mitigates computational difficulty of the enumeration while maintaining the prediction performance, and 2) to identify important subgraphs that contributes to high interpretability. We empirically show that EIN has sufficiently high prediction performance compared with standard graph neural network models, and also, we show examples of post-hoc analysis based on the selected subgraphs.

## 1 Introduction

Graph-level prediction tasks, which take a graph as an input and predict a label for the entire graph, have been widely studied in the data-science community. It is known that the graph representation is an effective approach to a variety of structure data such as chemical compounds (Ralaivola et al., 2005; Faber et al., 2017), protein structures (Gligorijević et al., 2021), and inorganic crystal structures (Xie & Grossman, 2018; Louis et al., 2020). In a graph-level prediction task, substructures on the input graph, i.e., subgraphs, are often an important factor for the prediction and the analysis. For example, in a prediction of a property of chemical compounds, identifying small substructures of the molecules can be essential for both of improving prediction accuracy and obtaining an insight about the underlying chemical mechanism. Therefore, mining predictive subgraphs is a significant issue for graph-level prediction tasks, and further, those subgraphs can have a higher order dependency to the prediction that requests sufficient flexibility in the model. However, building a prediction method that satisfies these requirements is a still challenging problem (see § 3 for existing studies).

Our proposed method, called Exact subgraph Isomorphism Network (EIN), adaptively identifies predictive subgraphs based on which a neural network model can be simultaneously trained. The overview of EIN is shown in Fig. 1. Fig. 1 (a) illustrates the subgraph representation of EIN denoted as $\psi_H(G)$ which takes a non-zero value if the input graph $G$ contains a subgraph $H$ and takes 0 otherwise (i.e., it is based on the exact subgraph isomorphism). As shown in Fig. 1 (b), EIN can be seen as a neural network in which the candidates of the input features are all the subgraphs contained in the training dataset. Because of its exact subgraph isomorphism representation, this architecture can be highly discriminative about subgraph structures. However, since the number of the candidate subgraphs can be enormous, the naïve computation of this architecture is computationally intractable.

EIN consists of Graph Mining Layer (GML) and Feed Forward Network (FFN). GML implements a mechanism to select only a small number of subgraphs necessary for the prediction, achieved through the group-sparse regularization. The group sparse regularization (Yuan & Lin, 2006) is a well-known regularizer for the group-wise selection of variables. We regard a set of neural network weights from an input unit (corresponding to one subgraph) to the next layer as a group, by which the adaptive subgraph selection can be realized. Only the selected subgraphs are used in the subsequent FFN

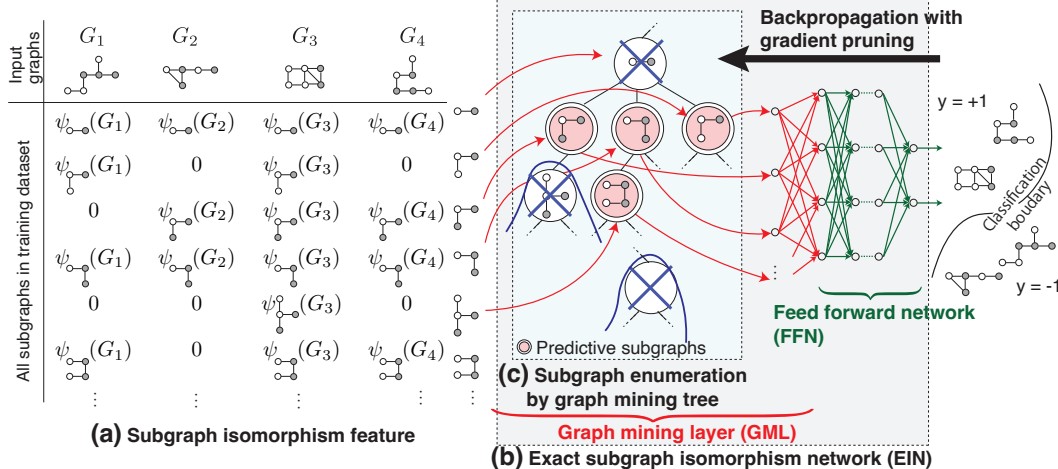

Figure 1: Overview of proposed method.

that learns the dependency between the selected subgraphs and the prediction target. Unfortunately, although the number of subgraphs becomes small in the trained model, this approach still suffers from the computational difficulty because all the candidate subgraphs are still required to consider during the optimization process.

To tackle the computational difficulty, we combine a subgraph enumeration by graph mining (Yan & Han, 2002) and the proximal gradient optimization (Teboulle, 2017; Beck & Teboulle, 2009), by which a pruning strategy for unnecessarily subgraphs can be derived, shown in 1 (c). The proximal gradient is a standard approach for the sparse modeling, in which parameters are updated through a proximal projection that typically results in a thresholding operation. This thresholding operation reveals that if the norm of the gradient corresponding to a subgraph $H$ is less than certain threshold, parameters for that subgraph is not required to update. We show that, by deriving an upper bound of the norm of the gradient, an efficient pruning strategy for the subgraph enumeration can be constructed. Our pruning strategy has the following two important implications. First, this pruning enables us to train EIN without enumerating all the candidate subgraphs, which makes EIN computationally tractable. Second, our pruning strategy maintains the quality of the prediction compared with when we do not perform the pruning. This is because we only omit the computations that does not have any effect on the prediction.

Our contributions are summarized as follows.

- We propose EIN, which is a neural network model that uses the exact subgraph isomorphism feature. Through a group sparse regularization, we formulate EIN so that a small number of subgraphs can be identified by which an insight about the important substructure can be extracted for the given graph-level prediction task.

- We show that the combination of the subgraph enumeration and the proximal gradient with backpropagation can derive an efficient pruning strategy that makes the EIN training computationally tractable. We further reveal that our pruning strategy does not degrade the prediction quality.

- Based on synthetic and benchmark datasets, we demonstrate that EIN has superior or comparable performance compared with standard graph prediction models while EIN actually can identify a small number of important subgraphs, simultaneously.

## 2 PROPOSED METHOD: EXACT SUBGRAPH ISOMORPHISM NETWORK

Our proposed method, called Exact subgraph Isomorphism Network (EIN), considers the classification problem of a graph $G \in \mathcal{G}$, where $\mathcal{G}$ is a set of labeled graphs. $G$ consists of a set of nodes and edges between nodes and each node can have a categorical label. The training data is $\{(G_i, y_i)\}_{i \in [n]}$,

where $y_i \in \mathcal{Y}$ is a graph label, $n$ is the number of instances, and $[n] = \{1, \ldots, n\}$. Here, although we only focus on the classification problem, the regression problem can also be handled by just replacing the loss function.

First, § 2.1 describes the formulation of our model. Second, the optimization procedure is shown in § 2.2. Next, § 2.3 and 2.4 discuss post-hoc analysis for knowledge discovery and a combination of graph neural networks, respectively.

## 2.1 MODEL DEFINITION

Let $\psi_H(G_i) \in \{0, 1\}$ be the feature that represents whether the input graph $G_i$ contains a subgraph $H$ (we only focus on a connected subgraph), which we call subgraph isomorphism feature (SIF):

$$\psi_H(G_i) = \mathbb{I}(H \sqsubseteq G_i), \tag{1}$$

where $\mathbb{I}$ is the indicator function and $H \sqsubseteq G_i$ indicates that $H$ is a subgraph of $G_i$. Note that although instead of $\mathbb{I}(H \sqsubseteq G_i)$, frequency that $H$ is included in $G_i$ can also be used for $\psi_H(G_i)$ in our framework, we employ (1) throughout the paper for simplicity (see Appendix A for more detail). For example, in Fig. 1 (a), $G_1$ is $\psi_H(G_1) = 1$ for $H = \circ\!\!-\!\!\bullet$ and $\psi_H(G_1) = 0$ for $H = \overset{\circ}{\underset{\circ}{\int}}\!\!\bullet$. We define candidate subgraphs as $\mathcal{H} = \{H \mid H \sqsubseteq G_i, i \in [n], |H| < \text{maxpat}\}$, which is all the subgraphs in the training dataset whose size is at most pre-specified maxpat (the size $|H|$ is the number of edges). By concatenating SIF $\psi_H(G_i)$ of each $H \in \mathcal{H}$, the feature vector $\boldsymbol{\psi}(G_i) \in \{0, 1\}^{|\mathcal{H}|}$ is defined. EIN identifies a small number of important subgraphs from $\mathcal{H}$ through the feature selection discussed later. SIF $\psi_H(G_i)$ is obviously highly interpretable and it can assure the existence of a subgraph $H$. Further, if $G_i$ and $G_j$ contain at least one different subgraph $H \in \mathcal{H}$, then, we have $\boldsymbol{\psi}(G_i) \neq \boldsymbol{\psi}(G_j)$.

As shown in Fig. 1 (b), EIN consists of Graph Mining Layer (GML) and Feed Forward Network (FFN). Let $K$ be the number of output units of GML. We define GML : $\mathcal{G} \rightarrow \mathbb{R}^K$ as follows.

$$\text{GML}(G_i; \boldsymbol{B}, \boldsymbol{b}) = \sigma(\boldsymbol{h}),$$

$$\boldsymbol{h} = \sum_{H \in \mathcal{H}} \boldsymbol{\beta}_H \psi_H(G_i) + \boldsymbol{b},$$

where $\boldsymbol{B} = [\boldsymbol{\beta}_1, \ldots, \boldsymbol{\beta}_{|\mathcal{H}|}] \in \mathbb{R}^{K \times |\mathcal{H}|}$ and $\boldsymbol{b} \in \mathbb{R}^K$ are parameters, and $\sigma : \mathbb{R}^K \rightarrow \mathbb{R}^K$ is an activation function. Each $\boldsymbol{\beta}_H$ can be seen as a representation corresponding to each subgraph $H$. The entire prediction model is defined as

$$f(G_i) = \text{FFN}(\text{GML}(G_i; \boldsymbol{B}, \boldsymbol{b}); \boldsymbol{\Theta}),$$

where $\boldsymbol{\Theta}$ is parameters of FFN. We optimize the parameters $\boldsymbol{B}, \boldsymbol{b}$, and $\boldsymbol{\Theta}$ through the following optimization problem in which the group sparse penalty (Yuan & Lin, 2006) is imposed on $\boldsymbol{\beta}_H$:

$$\min_{\boldsymbol{B}, \boldsymbol{b}, \boldsymbol{\Theta}} \sum_{i=1}^{n} \ell(y_i, f(G_i)) + \lambda \sum_{H \in \mathcal{H}} \|\boldsymbol{\beta}_H\|_2 \tag{2}$$

where $\ell$ is a differentiable loss function and $\lambda$ is a regularization parameter. We here use the cross-entropy loss for $\ell$. The group-wise penalty $\|\boldsymbol{\beta}_H\|_2$ results in $\boldsymbol{\beta}_H = \boldsymbol{0}$ for many unnecessarily $H$ at the solution of (2), by which we can identify important predictive subgraphs as $\{H \mid \boldsymbol{\beta}_H \neq \boldsymbol{0}\}$. However, since the size of $\boldsymbol{B}$, i.e., $K \times |\mathcal{H}|$, is quite large, naïve optimization of (2) can be difficult.

## 2.2 OPTIMIZATION

Our optimization algorithm is based on the block coordinate descent (Xu & Yin, 2017), in which each one of $\boldsymbol{B}, \boldsymbol{b}$, and $\boldsymbol{\Theta}$ are updated alternately while the other two parameters are fixed. Since $\boldsymbol{B}$ has the group sparse penalty, we apply the well-known proximal gradient method (Beck & Teboulle, 2009; Parikh & Boyd, 2014). On the other hand, $\boldsymbol{b}$ and $\boldsymbol{\Theta}$ have no sparse penalty and we can simply apply the standard gradient descent.

Let $\boldsymbol{g}_H = \frac{\partial}{\partial \boldsymbol{\beta}_H} \sum_{i=1}^{n} \ell(y_i, f(G_i))$ be the gradient of the loss function with respect to $\boldsymbol{\beta}_H$ and

$$\text{prox}(\boldsymbol{a}) = \begin{cases} \left(1 - \dfrac{\eta\lambda}{\|\boldsymbol{a}\|_2}\right)\boldsymbol{a} & \text{if } \|\boldsymbol{a}\|_2 > \eta\lambda, \\ \boldsymbol{0} & \text{if } \|\boldsymbol{a}\|_2 \leq \eta\lambda \end{cases} \tag{3}$$

be the proximal projection with respect to $\eta\lambda\|\cdot\|_2$ for a given vector $\boldsymbol{a}$ with the step size $\eta$. Then, the update of each parameter is defined as

$$\boldsymbol{\beta}_H^{(\text{new})} \leftarrow \text{prox}\,(\boldsymbol{\beta}_H - \eta\,\boldsymbol{g}_H) \text{ for } H \in \mathcal{H}, \tag{4}$$

$$\boldsymbol{b}^{(\text{new})} \leftarrow \boldsymbol{b} - \alpha \sum_{i=1}^{n} \frac{\partial \ell(y_i, f(G_i))}{\partial \boldsymbol{b}}, \tag{5}$$

$$\boldsymbol{\Theta}^{(\text{new})} \leftarrow \boldsymbol{\Theta} - \gamma \sum_{i=1}^{n} \frac{\partial \ell(y_i, f(G_i))}{\partial \boldsymbol{\Theta}}, \tag{6}$$

where $\alpha$ and $\gamma$ are step sizes. From the definition of (3), the proximal update (4) can be seen as a soft-thresholding that regularizes the standard gradient descent. Since $\mathcal{H}$ contains a large number of subgraphs, calculating (4) for all $\mathcal{H}$ is not directly tractable. In § 2.2.1, we derive a pruning rule by which we can perform (4) without enumerating all the subgraphs in $\mathcal{H}$. § 2.2.2 describes the entire procedure.

### 2.2.1 Backpropagation with Gradient Pruning

For the update of $\boldsymbol{\beta}_H$ defined as (4), we only update $H \in \mathcal{W}$ for a small working set $\mathcal{W} \subseteq \mathcal{H}$ instead of the entire $\mathcal{H}$. Let $\overline{\mathcal{W}} = \mathcal{H} \setminus \mathcal{W}$. Suppose that $\boldsymbol{\beta}_H$ is initialized as $\boldsymbol{0}$ and that $\boldsymbol{\beta}_H$ for $H \in \overline{\mathcal{W}}$ has never been updated, i.e., $\boldsymbol{\beta}_H = \boldsymbol{0}$ for $H \in \overline{\mathcal{W}}$. Then, from the proximal update rule (4), we see

$$\text{prox}\,(\boldsymbol{\beta}_H - \eta\,\boldsymbol{g}_H) = \boldsymbol{0} \text{ for } H \in \{H \mid \|\boldsymbol{g}_H\|_2 \leq \lambda \text{ and } H \in \overline{\mathcal{W}}\} \tag{7}$$

This means that if $\|\boldsymbol{g}_H\|_2 \leq \lambda$ and $H \in \overline{\mathcal{W}}$ are satisfied, both of the current $\boldsymbol{\beta}_H$ and the updated $\boldsymbol{\beta}_H^{\text{new}}$ are zero. From this observation, we incrementally update $\mathcal{W}$ as

$$\mathcal{W} \leftarrow \mathcal{W} \cup \left\{H \,\middle|\, \|\boldsymbol{g}_H\|_2 > \lambda, \forall H \in \overline{\mathcal{W}}\right\}, \tag{8}$$

and perform the update (4) only for $H \in \mathcal{W}$. However, evaluating $\|\boldsymbol{g}_H\|_2 > \lambda$ for all $H \in \overline{\mathcal{W}}$ is computationally demanding. To avoid this difficulty, the following theorem plays a key role:

**Theorem 2.1.** *Let $H' \sqsupseteq H$ for $H, H' \in \overline{\mathcal{W}}$, and $\delta_{ik} = \frac{\partial \ell(y_i, f(G_i))}{\partial h_k}$. Then, we have*

$$\|\boldsymbol{g}_{H'}\|_2 \leq \left\{\sum_{k=1}^{K} \max\left\{\Big(\sum_{i \in \{i \mid \delta_{ik} > 0\}} \delta_{ik}\psi_H(G_i)\Big)^2, \Big(\sum_{i \in \{i \mid \delta_{ik} < 0\}} \delta_{ik}\psi_H(G_i)\Big)^2\right\}\right\}^{1/2} =: \text{UB}(H). \tag{9}$$

We here only describe the sketch of the proof (the proof is in Appendix B). An essential idea is that we expand $\boldsymbol{g}_H$ as a linear combination of the derivatives with respect to the GML intermediate representation $h_k$, i.e., $\delta_{ik}$, by which the $k$-th element of the gradient can be written as $(\boldsymbol{g}_H)_k = \sum_{i=1}^{n} \delta_{ik}\psi_H(G_i)$. By using the monotonically non-increasing property of $\psi_H(G_i)$, i.e., $\psi_{H'}(G_i) \leq \psi_H(G_i)$ if $H \sqsubseteq H'$, we can derive $\text{UB}(H)$. Note that, usually, there is no need to explicitly consider this specific expansion of $(\boldsymbol{g}_H)_k$; the theorem can only be derived by deliberately reducing it to the linear combination of $\delta_{ik}$.

This theorem indicates that, for any $H'$ that contains $H$ as a subgraph, the L2 norm of the gradient $\|\boldsymbol{g}_{H'}\|_2$ can be bounded by $\text{UB}(H)$. Note that the upper bound $\text{UB}(H)$ can be calculated without generating $H'$. From the rule (7) and Theorem 2.1, we can immediately obtain the following important rule:

**Corollary 2.1.** *For $\forall H' \in \{H' \mid H' \sqsupseteq H, H' \in \overline{\mathcal{W}}\}$,*

$$\text{UB}(H) \leq \lambda \text{ and } H \in \overline{\mathcal{W}} \implies \text{prox}\,(\boldsymbol{\beta}_{H'} - \eta\,\boldsymbol{g}_{H'}) = \boldsymbol{0} \tag{10}$$

This is a direct consequence from the fact that if the conditions in the left side of (10) holds, we have $\|\boldsymbol{g}_{H'}\|_2 \leq \text{UB}(H) \leq \lambda$ from (9). As a result, by using (7), we obtain (10). Corollary (2.1) means that if the condition in (10) holds, for $\forall H'$ that contains $H$ as a subgraph, we can omit the update. Further, another notable advantage of this rule is that it does not depend on the step size $\eta$ (even when an iterative backtrack algorithm is employed for $\eta$, the evaluation of (10) is not required to repeat for every backtrack iteration).

By combining the rule (10) and a graph mining algorithm, we can build an efficient pruning algorithm. We employ a well-known graph mining algorithm called gSpan (Yan & Han, 2002) that can efficiently enumerate all the subgraphs in a given set of graphs. Figure 1 (c) is an illustration of a graph mining tree. At each tree node, gSpan expands the graph (add an edge and a node) as far as the expanded graph is included in the given dataset as a subgraph. By providing a unique code to each generated graphs (called the DFS code because it is based on depth-first search in a graph), gSpan can enumerate all the subgraphs without generating duplicated graphs (For further detail, see (Yan & Han, 2002)). As shown in Fig. 1 (c), all the graphs in the tree contains their ancestors as a subgraph. As a result, the following important consequence is obtained

**Remark 2.1.** *By evaluating* $\mathrm{UB}(H) \leq \lambda$ *during the tree traverse of gSpan (e.g., depth-first search), all the descendants of $H$ in the tree can be pruned if the inequality holds. This means that we can perform the update (8) without enumerating all the elements of $\overline{\mathcal{W}}$. In other words, we do not need to enumerate all the subgraphs $\mathcal{H}$ for (4).*

Further, another notable remark about the pruning is as follows.

**Remark 2.2.** *Since our pruning strategy only omit the update of unnecessarily parameters, i.e., $\beta_H = \mathbf{0}$, the pruning does no change all the parameter values compared with the case that we do not use the pruning. This also means that the final prediction performance also does not change because of the pruning.*

### 2.2.2 ALGORITHM

We here describe the optimization procedure of EIN. When the regularization parameter $\lambda$ is large, $\boldsymbol{B}$ becomes highly sparse, by which computations usually become faster because more subgraphs are expected to be pruned. Therefore, after starting from a larger value of $\lambda$, we gradually reduce $\lambda$ while optimizing parameters. Let $\Lambda = (\lambda_1, \ldots, \lambda_K)$ be a sequence of the regularization parameters, where $\lambda_1 > \lambda_2 \cdots > \lambda_K$. For each $\lambda$, we use the solution of previous $\lambda$ as an initial solution.

The Train-EIN function of Algorithm 1 performs the optimization (2) for each $\lambda$ from the given $\Lambda$. In line 5, the Traverse function performs the graph mining tree search in which $\mathcal{W}$ is generated via gradient pruning. The detailed procedure of the working set generation is in Algorithm 2. The Traverse function searches graph mining tree recursively. At each tree node, $\mathrm{UB}(H)$ is evaluated and the entire subtree below the current node can be pruned if $\mathrm{UB}(H) \leq \lambda$. On the other hand, if $\|\boldsymbol{g}_H\|_2 > \lambda$, $H$ should be included in $\mathcal{W}$. The children nodes of $H$ are created by gSpan in the Expand function, by which only the subgraphs $H'$ included in the training dataset can be generated and as a byproduct of this process, we obtain $\psi_{H'}(G_i)$. Note that if the children are already generated in the previous iterations, we can reuse them.

Once $\mathcal{W}$ is determined, $\boldsymbol{B}$ and $\boldsymbol{b}$ are updated in line 6-7 in Algorithm 1. Their step lengths are determined by backtrack algorithm (Note that since our pruning condition does not depend on the step length, we do not need to perform the Traverse function repeatedly during this backtrack steps). Then, $\boldsymbol{\Theta}$ is updated in line 8-10. This is a usual neural network parameter update, for which we

---

**Algorithm 1:** Optimization for EIN

1 **function** Train-EIN($\Lambda$)
2     $H_0 \leftarrow$ a graph at root of mining tree
3     **for** $\lambda$ in $\Lambda$ **do**
4         **repeat**
5             $\mathcal{W} \leftarrow$ Traverse($H_0, \mathcal{W}, \lambda$)
6             Update $\boldsymbol{B}$ by (4) for $H \in \mathcal{W}$
7             Update $\boldsymbol{b}$ by (5)
8             **for** iter $= 1, \ldots,$ MaxIter **do**
9                 Update $\boldsymbol{\Theta}$ by (6)
10             **end**
11         **until** terminate condition met
12     **end**
13 **end**

---

**Algorithm 2:** Working Set Generation

1 **function** Traverse($H, \mathcal{W}, \lambda$)
2     **if** $\mathrm{UB}(H) \leq \lambda$ **then return** $\mathcal{W}$
3     **if** $\|\boldsymbol{g}_H\|_2 > \lambda$ **then**
        $\mathcal{W} \leftarrow \mathcal{W} \cup \{H\}$
4     $C \leftarrow$ Expand($H$)
5     **for** $H' \in C$ **do**
6         $\mathcal{W} \leftarrow$ Traverse($H', \mathcal{W}, \lambda$)
7     **end**
8     **return** $\mathcal{W}$
9 **end**
10 **function** Expand($H$)
11     **if** children of $H$ have never been created by gSpan **then**
12         $C \leftarrow$ all graphs expanded from $H$ by gSpan
13         Set $\{\psi_{H'}(G_i)\}_{i=1}^n$ **for** $H' \in C$
14     **else**
15         $C \leftarrow$ Retrieve already expanded children of $H$
16     **end**
17     **return** $C$
18 **end**

---

iterate the update until given max iteration. Here, we employed the full instance gradient, the standard stochastic gradient can also be used. At each $\lambda$, the alternating update procedure stops (line 11) when the iteration reaches the given maximum iterations or some other stopping condition is satisfied.

### 2.3 Post-hoc Analysis for Knowledge Discovery

After the optimization, only a small number of $\beta_H$ have non-zero values, by which we can identify predictive subgraphs. Let $\mathcal{S} = \{H \in \mathcal{H} \mid \beta_H \neq \mathbf{0}\}$ be the set of the selected subgraphs by EIN. In our later experiments, we observed that $|\mathcal{S}|$ was typically ranged from around 50 to at most a few hundreds. Then, some insight may be obtained just by directly observing all of those selected subgraphs by the domain expert of the data.

The trained EIN can be seen as an $|\mathcal{S}|$-dimensional input neural network, to which post-hoc knowledge discovery methods (Bhati et al., 2025) can be applied. For example, well-known SHAP (Lundberg & Lee, 2017) and LIME (Ribeiro et al., 2016), both of which provide local feature importance, is applicable. Another typical approach is to use the trained EIN as a teacher and fit an interpretable surrogate model (e.g., decision tree), from which a possible decision rule can be estimated (Molnar, 2025). Note that, usually, applying interpretable machine learning methods directly to exhaustive subgraph isomorphism features can be computationally intractable. Because of EIN, which selects a few important subgraphs, a variety of post-hoc analyses become much easier.

### 2.4 Combining with Graph Neural Network

Since EIN is based on the standard backpropagation mechanism, general neural network models can be combined flexibly. For example, a simple approach to combining EIN with a graph neural network (GNN) is

$$f(G_i) = \text{FFN}(\text{GNN}(G_i; \boldsymbol{\Theta}_{\text{GNN}}) \\ \oplus \text{GML}(G_i; \boldsymbol{B}, \boldsymbol{b}); \boldsymbol{\Theta}_{\text{FFN}}), \quad (11)$$

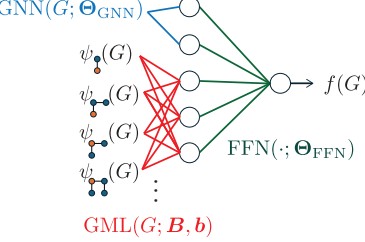

Figure 2: A simple example of EIN combined with GNN.

where $\oplus$ is the vector concatenation, GNN is any GNN that outputs arbitrary dimension representation vector, and $\boldsymbol{\Theta}_{\text{GNN}}$ and $\boldsymbol{\Theta}_{\text{FFN}}$ are parameters of GNN and FFN, respectively. This combination enables any GNNs to enhance the discriminative ability in terms of the selected subgraphs by EIN. A schematic illustration of a simplest case is shown in Fig. 2. Computations of the combined model can also be performed by almost the same alternating update (4)-(6). The only difference is that the FFN parameter update (6) is replaced with the simultaneous update of $\boldsymbol{\Theta}_{\text{FFN}}$ and $\boldsymbol{\Theta}_{\text{GNN}}$.

## 3 Related Work

For GNNs, the message passing based approach has been widely employed (Zhou et al., 2020). It is known that the expressive power of classical message passing approaches is limited by the first order Weisfeiler-Lehman (1-WL) test (Zhao et al., 2022). Many studies have tackled this limitation, some of which actually can be comparable with higher order WL tests. For example, $k$-hop extensions of the message passing (e.g., Abu-El-Haija et al., 2019; Nikolentzos et al., 2020; Wang et al., 2021; Chien et al., 2021; Brossard et al., 2020) and the higher order (tensor) representations (e.g., Morris et al., 2019; Maron et al., 2019b;a;c; Keriven & Peyré, 2019; Geerts & Reutter, 2022; Murphy et al., 2019) are popular approaches. Further, several studies (Bouritsas et al., 2023; Cotta et al., 2021; Barcelo et al., 2021; Bevilacqua et al., 2022) incorporate small subgraph information, such as pre-specified motifs and ego-networks, into neural networks. On the other hand, EIN performs the data-adaptive selection of a small number of (globally) predictive subgraphs, based on the exact enumeration of all the subgraphs in the dataset, which is obviously different approach from the above popular GNN studies. Further, most of GNNs can be combined with EIN by (11).

Although explainable GNNs are also studied, according to (Yuan et al., 2022), most of them are for the 'instance-level' explanation, which considers an explanation for the prediction of the specific one input graph. A few exceptions are approaches based on the maximization of the trained GNN output for a target label (Yuan et al., 2020), the latent space prototype learning (Azzolin et al., 2023),

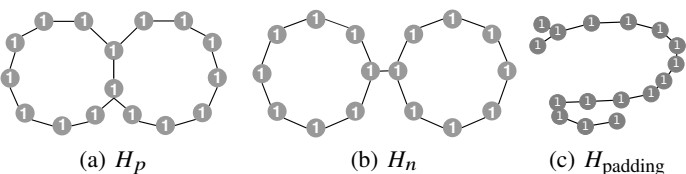

(a) $H_p$        (b) $H_n$        (c) $H_{\text{padding}}$

Figure 3: Subgraphs (a) and (b) have closed paths whose lengths are 8 and 9, respectively, which are difficult to discriminate. Subgraph (c) is used for adjusting the number of nodes that have label '1'. Note that all subgraphs (a), (b), and (c) have 16 nodes.

and the kernel-based filtering (Feng et al., 2022). However, unlike EIN, they cannot guarantee that the identified important graphs are actually subgraphs of the input graphs because none of them are based on the exact matching.

In the context of graph mining, discriminative pattern mining has also been studied (e.g., Thoma et al., 2010; Potin et al., 2025; Chen et al., 2022). However, most of them are not directly incorporated in a learning model. Typically, the subgraph enumeration (using some pruning strategies) is performed based on a discriminative score (e.g., ratio of frequency between two classes) independent from a learning model, and the selection results can be used in a subsequent prediction model training. Although (Nakagawa et al., 2016; Yoshida et al., 2019; 2021; Tajima et al., 2024) enumerate subgraphs during a model optimization, they are limited to a linear combination of the subgraphs. Note that Tajima et al. (2024) also use a combination of proximal gradient and gSpan, from which our optimization procedure is extended to the neural network backpropagation and the group-sparsity. Although (Nakagawa et al., 2016; Yoshida et al., 2019; 2021) provide a stronger pruning rule that can satisfy the global optimality, the convex formulation of the model training is required. To our knowledge, a discriminative subgraph mining that directly combines the exact subgraph enumeration with a (sparsity-induced) neural network has not been investigated.

## 4 EXPERIMENTS

We verify the prediction performance and interpretability of EIN through synthetic and benchmark datasets. For all the datasets, we partitioned them into train : valid : test = 6 : 2 : 2.

**Synthetic Datasets.** We created two synthetic datasets by using subgraphs shown in Figure 3. $H_p$ and $H_n$ are subgraphs with 16 nodes, which is known to be difficult to discriminate by standard GNNs.

The first dataset use $H_p$ and $H_n$, by which positive and negative classes are defined. We first generated a random connected graph, connected with one of $H_p$ or $H_n$ randomly. The initial random graph has node labels '0', and $H_p$ and $H_n$ have node labels '1'. We generated 300 instances for each of the positive and the negative classes. We call this dataset 'Cycle'. See Appendix C.1 for further details of the initial random graph.

In the second data, the positive and the negative classes are defined by the XOR rule, i.e., a nonlinear rule, of $H_p$ and $H_n$, which we call 'Cycle_XOR' By using the same random graph as the Cycle dataset, $H_p$ and/or $H_n$ are embedded into input graphs. Therefore, if a graph have one of $H_p$ or $H_n$, the class label is $y = 1$, otherwise $y = 0$, i.e., in the case that both of $H_p$ and $H_n$ are included or neither of them are included. We generated 150 instances each of four states of XOR, which results in 300 instances each of the positive and the negative classes. However, in this setting, the difference in the number of the nodes labeled as '1' may make the discrimination easier (32 if both of $H_p$ and $H_n$ exist, 16 if one of $H_p$ and $H_n$ exists, and 0 if neither of $H_p$ and $H_n$ exists). To avoid this, we added an simple subgraph $H_{\text{padding}}$ so that the number of the nodes labeled as '1' is 32 for all graphs.

**Benchmark Datasets.** As benchmark datasets, we used BZR, COX2, DHFR, and ENZYMES from (Morris et al., 2020), and ToxCast and SIDER from (Wu et al., 2018).

Table 1: Accuracy on synthetic and benchmark datasets

|  | BZR | COX2 | DHFR | ENZYMES | ToxCast | SIDER | Cycle | Cycle_XOR |
|---|---|---|---|---|---|---|---|---|
| GCN | 83.7±2.6 | 80.3±1.3 | 71.2±4.8 | 69.5±6.5 | 60.0±2.8 | 69.4±0.3 | 48.8±1.1 | 50.7±1.5 |
| GAT | 81.2±1.9 | 79.4±1.0 | 70.8±3.2 | 69.5±11.1 | 58.5±3.8 | 70.1±0.7 | 49.8±1.5 | 58.5±6.1 |
| GATv2 | 82.5±4.2 | 79.0±2.5 | 72.1±3.9 | 69.5±10.4 | 59.7±2.3 | 69.5±1.0 | 48.7±2.2 | 59.8±4.3 |
| GIN | 81.7±2.2 | 78.2±0.9 | 70.2±4.6 | 71.0±8.6 | 59.0±2.5 | 68.6±0.6 | 50.0±0.0 | 72.2±4.5 |
| PNA | 82.7±3.7 | 78.2±0.5 | 67.2±2.4 | 70.0±10.9 | 56.6±3.6 | 69.1±0.2 | 49.5±1.1 | 74.2±2.8 |
| GIN-AK | 82.5±2.3 | 80.5±3.0 | 77.1±3.2 | **75.5±8.0** | 58.7±2.5 | 68.9±1.7 | 53.2±1.6 | 72.0±5.4 |
| PPGN | 83.7±3.8 | 79.4±2.1 | 76.1±4.9 | 66.5±6.0 | 60.9±2.6 | 69.6±1.0 | 49.8±2.1 | 74.5±2.4 |
| **EIN** | **86.4±4.4** | **81.4±3.1** | **82.8±4.0** | 65.5±12.7 | 61.8±2.2 | 70.7±2.8 | **100.0±0.0** | **99.8±0.3** |
| **EIN+GIN** | 86.2±3.0 | 81.2±3.9 | 81.8±2.8 | 73.0±3.3 | **61.8±1.9** | 70.9±3.5 | **100.0±0.0** | 99.8±0.4 |
| # nonzero $\beta_H$ of EIN | 79±12 | 83±17 | 121±103 | 101±43 | 74±13 | 202±29 | 4.0±0.0 | 51±15 |

**Compared Methods.** For performance comparison, we used GNN methods such as GCN (Kipf & Welling, 2017), GAT (Velickovic et al., 2018), GATv2 (Brody et al., 2022), GIN (Xu et al., 2019), PNA (Corso et al., 2020), GNN-AK (Zhao et al., 2022), and PPGN (Maron et al., 2019a). For node attributes, EIN only used discrete node labels, while compared GNN methods also incorporate continuous node attributes. GCN, GAT, GATv2, and GIN optimizes the number of the units $\{64, 128, 256\}$ and the number of epoch by the validation set, and other settings follow (You et al., 2021). PNA optimizes the number of the units $\{16, 32, 64\}$ and the number of epochs by the validation set, and the number of the message passing is set as 2. GNN-AK uses GIN as the base GNN. Other settings of GNN-AK and PPGN follow the author implementation. For EIN, maxpat $\in \{5, 10\}$, $K \in \{2, 6, 10\}$, and $\sigma \in \{\text{sigmoid}, \text{LeakyReLU}\}$ are selected by the validation performance. We also evaluate performance of the combination of EIN and GNN described in § 2.4. For GNN, we employed GIN and the combine model is denoted as EIN+GIN. For other details, see Appendix C.2.

### 4.1 PREDICTION ACCURACY COMPARISON

Table 1 shows classification accuracy on synthetic and benchmark datasets. The results on Cycle and Cycle_XOR indicate that GNN based methods cannot discriminates closed paths shown in Fig. 3 (a) and (b). On the other hand, in Cycle and Cycle_XOR datasets, EIN and EIN+GIN achieved almost 100% classification. This indicates that EIN has a high discriminative ability about the subgraph structure, and flexibility that can capture a nonlinear relation.

For the other benchmark datasets, EIN and EIN+GIN show superior performance compared with other GNNs except for ENZYMES. EIN+GIN improves GIN for all datasets, which suggests that the exact subgraph information is essential for the prediction.

Further, the number of the selected subgraphs in EIN, shown in the bottom of Table 1 is at most a few hundreds, from which we see that EIN can effectively identify a small number of important subgraphs for the prediction.

### 4.2 EXAMPLES OF POST-HOC ANALYSIS

We here show examples of the post-hoc analysis using the trained EIN. Figure 4 is a result of SHAP for the EIN prediction on an instance of the ToxCast dataset. For SHAP, we used the python library `https://shap.readthedocs.io/en/latest/`, and see the document for detail. The SHAP values are for the predicted class (positive class), and in this case, we can see that a subgraph in Fig. 4 (a) has a particularly strong contribution to the prediction. Figure 5 shows an example of a fitted decision tree to the trained EIN prediction for the Cycle_XOR training and test datasets. In Fig. 5, the top node classifies a graph that do not have both of $H_p$ and $H_n$ as $y = 0$ (the top subgraph is included both in $H_p$ and $H_n$, but not in $H_{\text{padding}}$), and the second and the third nodes consists of the XOR rule by the length 8 and 9 closed paths (the second node is the length 9 closed path and the third node is the length 8 closed path). Additional examples are shown in Appendix D.

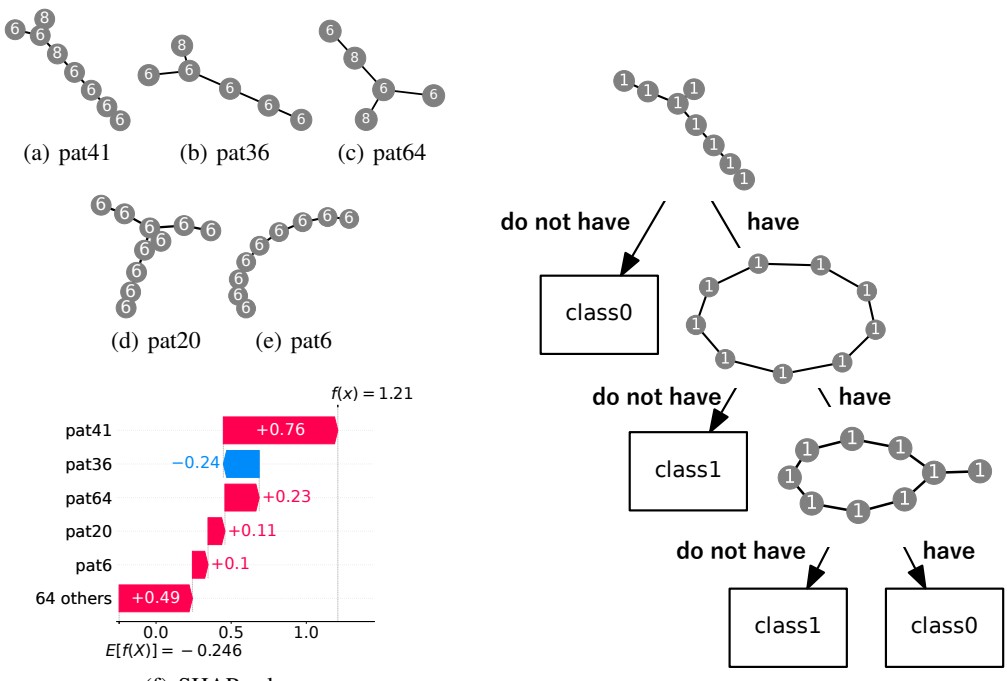

(a) pat41     (b) pat36     (c) pat64

(d) pat20     (e) pat6

(f) SHAP values

Figure 4: Example of SHAP applied to an EIN prediction from the ToxCast dataset.

Figure 5: Decision tree on Cycle_XOR

Table 2: Mean pruning rates of EIN

|  | BZR | COX2 | DHFR | ENZYMES | ToxCast | SIDER | Cycle | Cycle_XOR |
|---|---|---|---|---|---|---|---|---|
| Working set size | 76 | 146 | 91 | 164 | 60 | 193 | 4 | 50 |
| # Traverse nodes | 12637 | 5060 | 20596 | 45829 | 3710 | 6122 | 3 | 828 |
| # all subgraphs $|\mathcal{H}|$ | 112944 | 76120 | 108124 | 1133298 | 72734 | 101965 | 31953 | 40627 |
| Pruning rates (%) | 88.81 | 93.35 | 80.95 | 95.96 | 94.90 | 94.00 | 99.99 | 97.96 |

## 4.3 PRUNING RATES

Table 2 shows the mean working set size (mean over all $\lambda$), the mean number of the traversed nodes (mean over all $\lambda$), and the number of all the subgraphs $|\mathcal{H}|$ for maxpat = 10. The pruning rates are defined by the ratio between the number of the traversed nodes and $|\mathcal{H}|$. We can see that a large amount of nodes are pruned. Further, the mean size of the working set is at most a few hundreds, which means that $B$ is highly sparse during the optimization. Computational time required for the entire Algorithm 1 is also reported in Appendix E. The computational time of EIN is unfortunately higher than the standard GNNs. However, taking into account the fasts that EIN handles all the possible subgraphs with the exact matching and that the pruning rate was high, we consider that EIN performs highly efficiently computations. Further improvement for the high computational requirement is one of important future work to scale up the applicability.

## 5 CONCLUSIONS

We propose Exact subgraph Isomorphism Network (EIN). EIN combines the exact subgraph enumeration, neural networks, and the group sparse regularization. We show that predictive subgraphs can be identified efficiently by pruning unnecessarily subgraphs during the proximal update without sacrificing the quality of the model. We demonstrated that EIN has sufficiently high prediction accuracy compared with well-known graph neural networks despite that EIN only uses a small number of selected subgraphs.

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

## A  SIF by Frequency

The frequency based SIF is used in (Yoshida et al., 2021) in the context of a subgraph pattern based distance metric learning. They define $\#(H \sqsubseteq G_i)$ as the frequency of the subgraph $H$ contained in $G_i$, where nodes or edges among the counted subgraphs are not allowed. Then, the feature value is defined as

$$\phi_H(G_i) = g(\#(H \sqsubseteq G_i)),$$

where $g$ is a monotonically non-decreasing and non-negative function such as identity function $g(x) = x$ or the indicator function $g(x) = \mathbb{I}(\#(H \sqsubseteq G_i) > 0)$ (Yoshida et al. (2021) employed $g(x) = \log(1+x)$ for their evaluation). They pointed out computing the frequency without overlapping is NP-complete (Schreiber & Schwöbbermeyer, 2005), and approximate count is also provided (see (Yoshida et al., 2021) for detail). Our pruning theorem (Theorem 2.1) holds for both the exact $\phi_H(G_i)$ and its approximation.

## B  Proof of Theorem 2.1

We first transform the derivatives of the loss function with respect to $\boldsymbol{\beta}_H$ so that it is represented through the GML intermediate variable $\boldsymbol{h}$:

$$\boldsymbol{g}_H = \frac{\partial}{\partial \boldsymbol{\beta}_H} \sum_{i=1}^{n} \ell(y_i, f(G_i))$$

$$= \sum_{i=1}^{n} \frac{\partial \boldsymbol{h}^\top}{\partial \boldsymbol{\beta}_H} \frac{\partial \ell(y_i, f(G_i))}{\partial \boldsymbol{h}}.$$

From the definition of $\boldsymbol{h}$, we see $\frac{\partial \boldsymbol{h}^\top}{\partial \boldsymbol{\beta}_H} = \psi_H(G_i) \boldsymbol{I}_K$, where $\boldsymbol{I}_K$ is the $K \times K$ identity matrix. As a result, we have

$$\boldsymbol{g}_H = \sum_{i=1}^{n} \frac{\partial \ell(y_i, f(G_i))}{\partial \boldsymbol{h}} \psi_H(G_i).$$

Since $\delta_{ik} = \frac{\partial \ell(y_i, f(G_i))}{\partial h_k}$,

$$\|\boldsymbol{g}_{H'}\|_2 = \sqrt{\sum_{k=1}^{K} \left( \sum_{i=1}^{n} \delta_{ik} \psi_{H'}(G_i) \right)^2}. \tag{12}$$

From the definition, $0 \leq \psi_{H'}(G_i) \leq \psi_H(G_i)$. Then, the upper and lower bound of the inner sum of (12), i.e., $\sum_{i=1}^{n} \delta_{ik}\psi_{H'}(G_i)$, can be derived as

$$\sum_{i=1}^{n} \delta_{ik}\psi_{H'}(G_i) \leq \sum_{\{i|\delta_{ik}>0\}} \delta_{ik}\psi_{H'}(G_i) \leq \sum_{\{i|\delta_{ik}>0\}} \delta_{ik}\psi_H(G_i),$$

$$\sum_{i=1}^{n} \delta_{ik}\psi_{H'}(G_i) \geq \sum_{\{i|\delta_{ik}<0\}} \delta_{ik}\psi_{H'}(G_i) \geq \sum_{\{i|\delta_{ik}<0\}} \delta_{ik}\psi_H(G_i).$$

Therefore,

$$\left(\sum_{i=1}^{n} \delta_{ik}\psi_{H'}(G_i)\right)^2 \leq \max\left\{\left(\sum_{\{i|\delta_{ik}>0\}} \delta_{ik}\psi_H(G_i)\right)^2, \left(\sum_{\{i|\delta_{ik}<0\}} \delta_{ik}\psi_H(G_i)\right)^2\right\},$$

which results in (9).

## C  DETAIL OF EXPERIMENTAL SETTINGS

### C.1  SYNTHETIC DATASET

For both Cycle and Cycle_XOR, we first generate a random graph by the following procedure. The node size is randomly chosen from $5, 6, \ldots, 10$, and the number of edges are at most 20, which are also randomly generated by choosing node pairs uniformly. From $H_p$, $H_n$, or $H_{\text{padding}}$, randomly selected $n_{\text{connect}} \in \{3, 4, 5, 6\}$ nodes are connected to a randomly selected node in the initial random graph.

### C.2  OTHER SETTINGS OF EIN

In Algorithm 1, we selected five values of $\lambda$ taken from the interval $[\log(\lambda_{\max}), \log(0.01\lambda_{\max})]$. From $\log(\lambda_{\max})$, we iteratively decrease $\Delta\lambda$ five times (i.e., $\log(\lambda_{k+1}) = \log(\lambda_k) - \Delta\lambda$). The initial value of $\Delta\lambda$ is $(\log(\lambda_{\max}) - \log(0.01\lambda_{\max}))/5$. If the number of non-zero $\|\beta_H\|$ increase $\geq 10$, then we update $\Delta\lambda \leftarrow 0.5\Delta\lambda$. We determined $\lambda_{\max}$ by $\lambda_{\max} = \max_{H \in \mathcal{H}} \|g_H\|_2$, in which $B$ and $b$ were initialized by a linear model (Nakagawa et al., 2016) and $\Theta$ was randomly initialized. We set the terminate condition in the alternating update of Algorithm 1 as that the validation loss does not improve 5 times. The hyper-parameters $\lambda$, maxpat $\in \{5, 10\}$, $K \in \{2, 6, 10\}$, and $\sigma \in \{\text{sigmoid}, \text{LeakyReLU}\}$ were selected by the validation loss. We set the number of layers of FFN as 1 and MaxIter for $\Theta$ was 30.

In EIN+GIN, the message passing in GIN was set as 3, the number of the middle unit was 16, and the activation function was ReLU.

## D  OTHER EXAMPLES OF POST-HOC ANALYSIS

Figure 6 shows an example of a fitted decision tree to the trained EIN prediction for the SIDER training and test datasets, which provides possible decision rule behind subgraphs and the target label. As another simple example of post-hoc analysis, Fig. 7 shows subgraph importance estimated by Random Forest (RF) (Breiman, 2001) for Cycle_XOR. We fitted RF to a set of $(\psi_S(G_i), f(G_i))$ created by the training and test datasets and the importance is evaluated by `scikit-learn` feature importance of `RandomForestRegressor`, which is based on mean decrease of impurity by that feature. Top five important subgraphs are shown in the figures. Figure 7 indicates that the closed paths of $H_p$ and $H_n$ are identified by EIN (the top subgraph is the length 9 closed path and the second subgraph is the length 8 closed path).

## E  COMPUTATIONAL TIMES

The computational time spent on Algorithm 1 is shown in Table 3. In the table, 'Traverse Times' corresponds to the Traverse function in Algorithm 1. We see that the time required for the subgraph

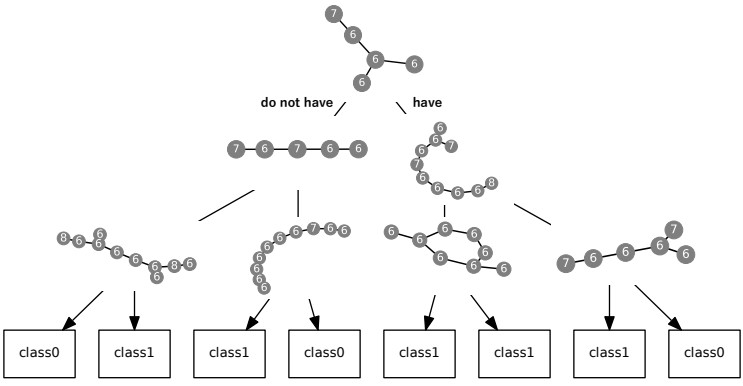

Figure 6: Decision tree on SIDER.

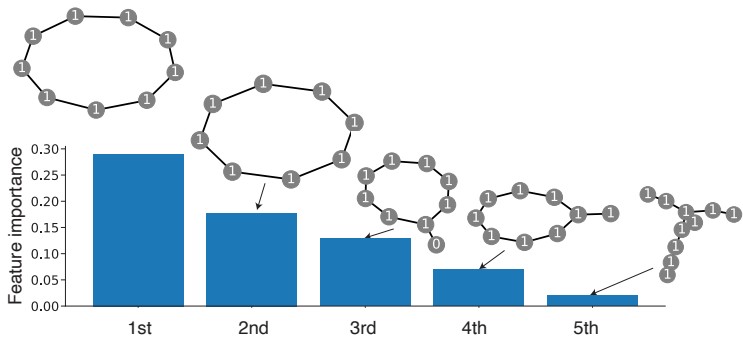

Figure 7: Subgraph importance estimated by RF on Cycle_XOR dataset.

enumerations depends on the datasets (depends on a variety of factors such as node sizes, edge sizes, and the pruning rate).

Table 3: Mean computational time for the entire Algorithm 1

|  | BZR | COX2 | DHFR | ENZYMES | ToxCast | SIDER | Cycle89 | Cycle89_XOR |
|---|---|---|---|---|---|---|---|---|
| Traverse Times (s) | 13 | 9.2 | 68 | 991 | 602 | 1706 | 85 | 7692 |
| All times (s) | 179 | 185 | 598 | 1089 | 744 | 1859 | 1253 | 10361 |

## F LLM USAGE

In this manuscript, LLM was only used to polish writing.

