# OpenReview forum: "Exact Subgraph Isomorphism Network for Predictive Graph Mining"
_ICLR.cc/2026/Conference — ICLR 2026 Conference Withdrawn Submission_

### Official Review · Reviewer_cxww · 2025-11-01

**Soundness:** 4
**Presentation:** 3
**Contribution:** 3
**Rating:** 4
**Confidence:** 4

**Summary:**

The paper proposes EIN, a graph-level prediction model that (i) represents each training graph by exact subgraph-isomorphism features (SIF) over all subgraphs up to size maxpat, (ii) learns a small, task-specific set of predictive subgraphs via group-sparse (ℓ₂,₁) regularization in a Graph Mining Layer (GML), and (iii) trains an FFN head on the selected SIF. To make the subgraph search feasible, it couples gSpan enumeration with a proximal-gradient–driven pruning rule that bounds per-subgraph gradient norms and prunes entire subtrees of the mining tree; empirically, the method yields competitive accuracy and identifies a compact set of subgraphs that support post-hoc interpretability.

**Strengths:**

- Exact, globally interpretable features. Using exact subgraph matches (not counts/heuristics) provides crisp semantics. Feature selection via group sparse yields a small, human-auditable subgraph set.
- Clear algorithmic presentation. The optimization loop and working-set traversal are explicit.

**Weaknesses:**

- Scalability and cost remain open. Even with pruning, the approach requires on-the-fly SIF generation/caching and many gSpan traversals. Stronger time/memory breakdowns and scale-up curves would help.
- Baseline breadth. The main tables compare to GCN/GAT/GIN/PNA/GNN-AK/PPGN, but missing subgraph-aware SOTA baselines.
- Number of total subgraphs scales exponentially with the number of nodes, which means the proposed method might be intractable for larger graphs. This could limit the impact of the paper.
- The reviwer is concerning the expressivity of the proposed method. Since the size of subgraph is controled by 'maxpat', there could be some graphs that have the same SIF. For example, two cycles with length n and 2n ('maxpat'<n). Then, these two cycles will have the same SIF but they are not isomorphic. A ablation on 'maxpat' would help.

**Questions:**

- Can the author explain, instead of l-1 nrom, why l-2 norm penalty is used to enforce sparsity?

---

### Official Review · Reviewer_gLDX · 2025-11-01

**Soundness:** 3
**Presentation:** 3
**Contribution:** 1
**Rating:** 2
**Confidence:** 3

**Summary:**

The paper introduces an Exact Subgraph Isomorphism Network (EIN) for the graph classification task. The authors proposed to use gSpan — a graph-mining algorithm — to construct a diverse set of subgraphs and use the membership of subgraphs within each graph as isomorphic features. To reduce the number of total subgraphs, a gradient pruning technique based on the gradient bound is proposed for adaptive subgraphs selection. EIN shows superior accuracy on synthetic datasets and competitive performance to GNN baselines on different real-world datasets, with added interpretability via post-hoc analysis.

**Strengths:**

- The paper is well-written, ideas are clearly presented and easy to follow.
- The pruning rule is theoretically grounded and preserves prediction quality.
- Experiments demonstrate effectiveness on hard synthetic datasets (e.g., Cycle and Cycle_XOR, where EIN achieves ~100% accuracy vs. GNNs' ~50-70%), with clear ablation on pruning rates.
- The subgraph selection method enables post-hoc analysis which enhances the interpretability of the model.

**Weaknesses:**

- Given previous work [1], in which the the membership score and the gradient pruning technique are proposed, I found the technical contribution of this work is not significant, which only adds a non-linear layer for the prediction model.​
- Computational times could be exponentially scaled with number of nodes, and maxpat=10 limits applicability to larger graphs/subgraphs, broader scaling analysis and ablation on maxpat are needed.
- The pruning based on the gradient of the lost w.r.t. the parameters, which is based on the current model’s knowledge of the data, and could be biased in early steps.
- The code is not provided and the details of selected parameters are missing limit the application/reproducibility of the paper.

[1] Tajima, Shinji, et al. "Learning Attributed Graphlets: Predictive Graph Mining by Graphlets with Trainable Attribute." Proceedings of the 30th ACM SIGKDD Conference on Knowledge Discovery and Data Mining. 2024.

**Questions:**

- A more detail comments on the differences against [1] is needed.
- Analysis on the expressiveness of the model is highly recommended, it is not clear to me how expressive the proposed model is compared with other baseline GNNs.

---

### Official Review · Reviewer_9Re5 · 2025-11-01

**Soundness:** 3
**Presentation:** 3
**Contribution:** 3
**Rating:** 4
**Confidence:** 1

**Summary:**

The authors propose a graph representation learning framework that performs graph classification by leveraging the exact subgraph isomorphism features. This work groups sparse weights over all candidate subgraphs and captures interactions among sparsely selected subgraphs and bounds gradients during backpropagation to prune tree branches for tractable training. Experimental results show that EIN outperforms baselines for graph classification tasks.

**Strengths:**

· This work has great theoretical insight that showed exact isomorphism exceeds 1-WL expressivity.

· The authors provide extensive reproducible codes to replicate their experiments.

· The method is cleanly formalized with complete pseudocode using standard tools (gSpan, proximal gradient, backtracking).

**Weaknesses:**

· The experiment section in this paper does not have any recent baselines. The most recent baseline used in this paper is from 2019, while newer works exist that need to be used as baselines [1] [2] [3] [4] [5].

· The ablation studies are minor (frequency vs binary, deeper FFN, etc.)

· The entire framework seems to be dependent on the maxpat hyperparameter (maximum subgraph size). The computational complexity grows combinatorially with maxpat. The paper uses maxpat=10, but many meaningful substructures in domains like chemistry (e.g., functional groups) or social networks (e.g., small communities) might require larger patterns to be discriminative. The method provides no scalable path to discover these.

· Theorem 2.1 shows that the effectiveness of the pruning hinges on the UB(H) being reasonably tight. If UB(H) is a loose over-estimate, the pruning will be far less effective than claimed, as many branches of the gSpan tree that should be pruned

will not be. The paper provides no analysis or empirical evidence regarding the tightness of this bound across different datasets.

· The results for EIN+GIN are presented, but the analysis is shallow. It is unclear how the GNN and EIN components interact. Does the GNN learn complementary features that the subgraph features miss? Does the presence of the GNN change which subgraphs are selected by EIN? A deeper analysis of this interaction is needed to justify the combined architecture.

· The method is fundamentally limited by the subgraph enumeration process. Even with a high pruning rate, the number of traversed nodes and the total computation time are immense. For example, the Cycle_XOR dataset, which only has 600 graphs, required over 10000s to train. This is orders of magnitude slower than standard GNNs. The approach does not scale to large-scale graph datasets like those in molecular biology or social networks, where graphs can have hundreds of nodes.



[1] Yu, Zhaoning, and Hongyang Gao. "Molecular representation learning via heterogeneous motif graph neural networks." International conference on machine learning. PMLR, 2022.

[2] Yan, Zuoyu, et al. "An efficient subgraph gnn with provable substructure counting power." Proceedings of the 30th ACM SIGKDD Conference on Knowledge Discovery and Data Mining. 2024.

[3] Chen, Kaixuan, et al. "Improving expressivity of gnns with subgraph-specific factor embedded normalization." Proceedings of the 29th ACM SIGKDD Conference on Knowledge Discovery and Data Mining. 2023.

[4] Zhang, Muhan, and Pan Li. "Nested graph neural networks." Advances in Neural Information Processing Systems 34 (2021): 15734-15747.

[5] Zhao, Lingxiao, et al. "From stars to subgraphs: Uplifting any GNN with local structure awareness." arXiv preprint arXiv:2110.03753 (2021).

**Questions:**

Please address the weaknesses listed above.

---

### Official Review · Reviewer_kTFE · 2025-11-05

**Soundness:** 3
**Presentation:** 2
**Contribution:** 3
**Rating:** 6
**Confidence:** 4

**Summary:**

This paper proposes the Exact subgraph Isomorphism Network (EIN), a framework for graph-level prediction. EIN treats the 0/1 existence of all connected subgraphs found in the training dataset up to a certain size as a high-dimensional feature set.  It then uses an FFN to classify the features. The claimed key novelty is in the optimization of the framework which allows for significant pruning in gSpan-based subgraph enumeration, due to a derived gradient norm upper bound. The authors demonstrate EIN's effectiveness on various synthetic and real-world datasets.

**Strengths:**

1. This is an interesting paper that re-invents a simple (yet effective) idea of using subgraph mining to do graph-level prediction. For a long time the bottleneck of executing this idea in practice is in subgraph enumeration. The authors innovatively and rigorously show that with proximal gradient descent and sparse regularization, we can have a nice optimization form that prunes the vast majority of subgraphs in the universe.
2. The studied topic is certainly of importance, and empirical performance and analysis are solid.

**Weaknesses:**

1. It seems to me that the the core theorem (Thm 2.1 and Coro. 2.1) may not hold if the feature is generalized to the count of subgraps rather than the 0/1 existence. If the theorems still hold, it would be very helpful to show it. If the theorem doesn't hold, I think working with only the count of subgraphs would not give the best expressiveness.
2. It would be helpful to provide sensitivity analysis of  "maxpat" which is an important hyperparameter.
3. Minor: "for many unnecessarily 𝐻" in Line 148 should be  "for many unnecessary 𝐻"; Figure 1 should have a caption that illustrates the method in place.

**Questions:**

See weaknesses.

---

### Note · Authors · 2025-12-03

I have read and agree with the venue's withdrawal policy on behalf of myself and my co-authors.